# Transfer Learning for Adenocarcinoma Classifications in the Transurethral Resection of Prostate Whole-Slide Images

**DOI:** 10.3390/cancers14194744

**Published:** 2022-09-28

**Authors:** Masayuki Tsuneki, Makoto Abe, Fahdi Kanavati

**Affiliations:** 1Medmain Research, Medmain Inc., Fukuoka 810-0042, Japan; 2Department of Pathology, Tochigi Cancer Center, 4-9-13 Yohnan, Utsunomiya 320-0834, Japan

**Keywords:** weakly supervised learning, transfer learning, deep learning, adenocarcinoma, transurethral resection of the prostate, whole-slide image

## Abstract

**Simple Summary:**

In this study, we trained deep learning models to classify TUR-P WSIs into prostate adenocarcinoma and benign (non-neoplastic) lesions using transfer and weakly supervised learning. Overall, the model achieved good classification performance in classifying whole-slide images, demonstrating the potential benefit of future deployments in a practical TUR-P histopathological diagnostic workflow system.

**Abstract:**

The transurethral resection of the prostate (TUR-P) is an option for benign prostatic diseases, especially nodular hyperplasia patients who have moderate to severe urinary problems that have not responded to medication. Importantly, incidental prostate cancer is diagnosed at the time of TUR-P for benign prostatic disease. TUR-P specimens contain a large number of fragmented prostate tissues; this makes them time consuming to examine for pathologists as they have to check each fragment one by one. In this study, we trained deep learning models to classify TUR-P WSIs into prostate adenocarcinoma and benign (non-neoplastic) lesions using transfer and weakly supervised learning. We evaluated the models on TUR-P, needle biopsy, and The Cancer Genome Atlas (TCGA) public dataset test sets, achieving an ROC-AUC up to 0.984 in TUR-P test sets for adenocarcinoma. The results demonstrate the promising potential of deployment in a practical TUR-P histopathological diagnostic workflow system to improve the efficiency of pathologists.

## 1. Introduction

According to the Global Cancer Statistics 2020, prostate cancer is the most frequently diagnosed cancer in men in over one-half (112 of 185) of the countries in the world. It is the fifth leading cause of cancer death among men in 2020 with an estimated 1,414,259 new cases and 375,304 deaths worldwide [1]. The only way to properly diagnose prostate cancer is via histopathological confirmation [2].

Nodular hyperplasia (benign prostatic hyperplasia) is a common benign disorder of the prostate that represents a nodular enlargement of the gland caused by hyperplasia of both glandular and stromal components, resulting in an increase in the weight of the prostate. The conventional treatment for nodular hyperplasia is surgical. The transurethral resection of the prostate (TUR-P) is one of the most widely practiced surgical procedures, which is estimated to have been performed in about 7864 cases in 2014 in Japan [3]. TUR-P can be used as an incidental diagnosis for prostate cancer with the gold standard for prostate cancer being a biopsy. In TUR-P, an electrical loop of resectoscope excises hyperplastic prostate tissues to improve urine flow, resulting in many tiny tissue fragments with variable sizes during the procedure. As compared with conventional biopsy specimens (e.g., endoscopic biopsy from gastrointestinal tracts), TUR-P specimens are characterized by a very large volume of tissues and a large number of glass slides; therefore, the histopathological diagnosis for TUR-P specimen is one of the most tedious and error-prone tasks because there are a large number of tissue artifacts, and determining the orientation of the specimen is difficult. Importantly, cancers, especially prostate adenocarcinoma, are detected incidentally around 5–17% of TUR-P specimens [4,5,6,7,8,9,10]. Conventional active treatments (surgery or radiotherapy) are indicated in T1a patients with a life expectancy that is longer than 10 years and in the majority of T1b patients [4]. The histopathological evaluation of cancer (adenocarcinoma) in TUR-P specimens is important because the presence of cancer in more than 5% of the tissue fragments [11,12] or high-grade cancer [13,14] may affect the choice of treatment. Thus, for TUR-P specimens, reporting both the number of microscopic foci of carcinoma and the percentage of carcinomatous involvement is recommended. All these factors and burdens mentioned above highlight the benefit of establishing a histopathological screening system to detect prostate adenocarcinoma based on TUR-P specimens. Conventional glass slides of TUR-P specimens can be digitized as whole-slide images (WSIs), which could benefit from the application of computational histopathology algorithms, especially deep learning models, for aiding pathologists, reducing the burden of time-consuming diagnosis, and increasing the appropriate detection rate of prostate adenocarcinoma in TUR-P WSIs as part of a screening system.

In computational pathology, deep learning models have been widely applied in histopathological cancer classification on WSIs, cancer cell detection and segmentation, and the stratification of patient outcomes [15,16,17,18,19,20,21,22,23,24,25,26,27,28]. Previous studies have looked into applying deep learning models for adenocarcinoma classification in stomach [28,29,30], colon [28,31], lung [29,32], and breast [33,34] histopathological specimen WSIs. In a previous study, we trained a prostate adenocarcinoma classification model on needle biopsy WSIs [35] and evaluated the models on both needle biopsy and TUR-P WSI test sets to confirm their applications in different types of specimens, achieving an ROC-AUC of up to 0.978 in needle biopsy test sets; however, the model under-performed on TUR-P WSIs. Therefore, in this study, we trained deep learning models specifically for TUR-P WSIs. We evaluated the trained models on TUR-P, needle biopsy, and TCGA (The Cancer Genome Atlas) public dataset test sets, achieving an ROC-AUC of up to 0.984 in TUR-P test sets, 0.913 in needle biopsy test sets, and 0.947 in TCGA public dataset test sets. These findings suggest that deep learning models might be very useful as routine histopathological diagnostic aids for inspecting TUR-P WSIs to detect prostate adenocarcinoma precisely.

## 2. Materials and Methods

### 2.1. Clinical Cases and Pathological Records

Retrospectively, a total of 2060 H&E (hematoxylin & eosin)-stained histopathological specimen slides of human prostate adenocarcinoma and benign (non-neoplastic) lesions were collected from the surgical pathology files of a total of three hospitals, Shinyukuhashi, Wajiro, and Shinkuki hospitals (Kamachi Group Hospitals, Fukuoka, Japan), after a histopathological review of all specimens by surgical pathologists. Of the 2060 slides, 1560 were TUR-P obtained from 276 patients and 500 were needle biopsy obtained from 238 patients. All were obtained between 2017 and 2019. Histopathological specimens were selected randomly to reflect real clinical settings as much as possible. Prior to the experimental procedures, each WSI diagnosis was observed by at least two pathologists (at least over five years experience) with the final checking and verification performed by senior pathologists (at least over 10 years experience). The pathologists had to agree whether the output was adenomcarinoma or benign. All WSIs were scanned at a magnification of ×20 using the same Leica Aperio AT2 Digital Whole Slide Scanner (Leica Biosystems, Tokyo, Japan) and were saved in the SVS file format with JPEG2000 compression.

### 2.2. Dataset

Hospitals that provided histopathological specimen slides in the present study were anonymised (e.g., Hospital-A, B, and C). Table 1 breaks down the distribution of training and validation sets of TUR-P WSIs from two domestic hospitals (Hospital-A and B). Validation sets were selected randomly from the training sets (Table 1). The test sets consisted of TUR-P, needle biopsy, and the TCGA (The Cancer Genome Atlas) public dataset WSIs (Table 2). The distribution of test sets from three domestic hospitals (Hospital-A, B, and C) and the TCGA public dataset is summarized in Table 2. Patients’ pathological records were used to extract WSIs’ pathological diagnoses and to assign WSI labels. Training set WSIs were not annotated, and the training algorithm only used WSI diagnosis labels, meaning that the only information available for training was whether the WSI contained adenocarcinoma or was benign (non-neoplastic lesion), but no information about the location of the cancerous tissue lesions was provided. External prostate TCGA datasets are publicly available from the Genomic Data Commons (GDC) Data Portal (https://portal.gdc.cancer.gov/, accessed on 18 January 2022). We have confirmed that surgical pathologists were able to diagnose test sets in Table 2 from the visual inspection of the H&E-stained WSIs slide alone.

### 2.3. Deep Learning Models

We trained the models via transfer learning using the partial finetuning approach [36]. This is an efficient fine-tuning approach that consists of using the weights of an existing pre-trained model and only finetuning the affine parameters of batch-normalization layers and the final classification layer. For the model’s architecture, we used EfficientNetB1 [37] starting with pre-trained weights on ImageNet. Figure 1 shows an overview of the training method. The training methodology that we used in the present study was exactly the same as reported in our previous studies [29,35]. For the sake of completeness, we repeat the methodology here.

We performed slide tiling by extracting square tiles from tissue regions of the WSIs. We started by detecting the tissue regions in order to eliminate most of the white background. This was conducted by performing thresholding on a grayscale version of the WSIs using Otsu’s method [38]. During prediction, we performed the tiling of the tissue regions in a sliding window fashion by using a fixed-size stride (256 × 256 pixels). During training, we initially performed the random balanced sampling of tiles extracted from tissue regions, where we tried to maintain an equal balance of each label in the training batch. To do so, we placed WSIs in a shuffled queue such that we looped over the labels in succession (i.e., we alternated between picking a WSI with a positive label and a negative label). Once a WSI was selected, we randomly sampled $\frac{\mathrm{batch}\;\mathrm{size}}{\mathrm{num}\;\mathrm{labels}}$ tiles from each WSI to form a balanced batch.

To maintain the balance on the WSI, we oversampled from WSIs to ensure that the model trained on tiles from all WSIs in each epoch. We then switched to hard mining tiles. To perform hard mining, we alternated between training and inference. During inference, the CNN was applied in a sliding window fashion on all the tissue regions in the WSI, and we then selected the *k* tiles with the highest probability for being positive. This step effectively selects tiles that are most likely to be false positives when the WSI is negative. The selected tiles were placed in a training subset, and once that subset contained *N* tiles, training was initiated. We used k=8, N=256, and a batch size of 32.

To obtain a single prediction for the WSIs from tile predictions, we took the maximum probability from all tiles. We used the Adam optimizer [39], with the binary cross-entropy as the loss function and with the following parameters: $beta_{1}=0.9$, $beta_{2}=0.999$, a batch size of 32, and a learning rate of 0.001 when finetuning. We used early stopping by tracking the performance of the model on a validation set, and training stopped automatically when there were no further improvements on the validation loss for 10 epochs. We chose the model with the lowest validation loss as the final model.

### 2.4. Software and Statistical Analysis

Deep learning models were implemented and trained using TensorFlow [40]. AUCs were calculated in Python using the scikit-learn package [41] and plotted using matplotlib [42]. 95% CIs of the AUCs were estimated using the bootstrap method [43] with 1000 iterations.

The true positive rate (TPR) (also called sensitivity) was computed as follows.
(1)$$TPR=\frac{TP}{TP+FN}$$

The false positive rate (FPR) was computed as follows.
(2)$$FPR=\frac{FP}{FP+TN}$$

The true negative rate (TNR) (also called specificity) was computed as follows:(3)$$TNR=\frac{TN}{FP+TN}$$
where TP, FP, and TN represent true positive, false positive, and true negative, respectively. The ROC curve was computed by varying the probability threshold from 0.0 to 1.0 and computing both the TPR and FPR at the given threshold.

## 3. Results

### 3.1. Insufficient AUC Performance of WSI Prostate Adenocarcinoma Evaluation on TUR-P WSIS Using Existing Series of Adenocarcinoma Classification Models

Prior to training a new prostate adenocarcinoma model using TUR-P WSIs, we applied existing adenocarcinoma classification models and evaluated their AUC performances on TUR-P test sets (Table 2). Existing adenocarcinoma classification models were summarized in Table 3: (1) breast invasive ductal carcinoma (IDC) classification model (Breast IDC (×10, 512)) [33]; (2) breast invasive ductal carcinoma and ductal carcinoma in situ (DCIS) classification model (Breast IDC, DCIS (×10, 224)) [34]; (3) colon adenocarcinoma (ADC) and adenoma (AD) classification model (Colon ADC, AD (×10, 512)) [28]; (4) colon poorly differentiated adenocarcinoma classification model (transfer learning model from stomach poorly differentiated adenocarcinoma classification model) (Colon poorly ADC-1 (×20, 512)) [31]; (5) colon poorly differentiated adenocarcinoma classification model (EfficientNetB1 trained model) (Colon poorly ADC-2 (×20, 512)) [31]; (6) stomach adenocarcinoma and adenoma classification model (Stomach ADC, AD (×10, 512)) [28]; (7) stomach poorly differentiated adenocarcinoma classification model (Stomach poorly ADC (×20, 224)) [29]; (8) stomach signet ring cell carcinoma (SRCC) classification model (Stomach SRCC (×10, 224)) [30]; (9) pancreas endoscopic ultrasound-guided fine needle aspiration (EUS-FNA) biopsy adenocarcinoma classification model (Pancreas EUS-FNA ADC (×10, 224)) [44]; and (10) lung carcinoma classification model (Lung Carcinoma (×10, 512)) [45]. Table 3 shows that Colon poorly ADC-2 (×20, 512) and Lung Carcinoma (×10, 512) models exhibited both high ROC-AUC and low log loss values compared to other models. Thus, we used the Colon poorly ADC-2 (×20, 512) and Lung Carcinoma (×10, 512) models as initial weights for finetuning on the TUR-P training sets when performing transfer learning (Table 1).

### 3.2. High AUC Performance of TUR-P WSI Evaluation of Prostate Adenocarcinoma Histopathology Images

We trained models using transfer learning (TL) and weakly supervised learning approaches, which allow the use of weak labels (WSI labels) [35,45]. These models are all based on the EfficientNetB1 convolutional neural network (CNN) architecture. For comparison, we also trained two models using the EfficientNetB1 architecture at a magnification of ×10 and ×20 using initial weight training on ImageNet. The models were applied in a sliding window fashion with input tiles of 224 × 224 and 512 × 512 pixels and a stride of 256 (Figure 1). To train deep learning models, we used a total of 79 adenocarcinoma and 941 benign training set WSIs and 20 adenocarcinoma and 20 benign validation set WSIs (Table 1). This resulted in four different models: (1) TL-colon poorly ADC-2 (×20, 512), (2) TL-lung carcinoma (×10, 512), (3) EfficientNetB1 (×10, 224), and (4) EfficientNetB1 (×20, 512). We evaluated four different trained deep learning models on test sets from three different hospitals (Hospital-A–C) and TCGA public datasets (Table 2). For each test set (TUR-P: Hospital-A–B, TUR-P: Hospital-A, TUR-P: Hospital-B, public dataset: TCGA, and needle biopsy: Hospital-A–C), we computed the ROC-AUC, log loss, accuracy, sensitivity, and specificity and them summarized in Table 4 and Table 5 and Figure 2. The transfer learning model (TL-colon poorly ADC-2 (×20, 512)) (Figure 2A) from the existing colon poorly differentiated adenocarcinoma classification model (Colon poorly ADC-2 (×20, 512)) [31] trained using TUR-P training sets have higher ROC-AUCs and lower log losses compared to the other models (TL-lung carcinoma (×10, 512), EfficientNetB1 (×10, 224), and EfficientNetB1 (×20, 512)) (Table 4, Figure 2). On the other hand, on TUR-P hospital-B test sets, both EfficientNetB1 (×10, 224) and EfficientNetB1 (×20, 512) models exhibited very high ROC-AUCs (0.924–0.973) and low log-losses (0.126–0.251) compared to the other test sets (TUR-P Hospital-A, public dataset, and needle biopsy) (Table 4 and Figure 2C,D). Looking at heatmap images of the same TUR-P WSI that were correctly predicted as prostate adenocarcinoma using four different trained models, both EfficientNetB1 (×10, 224) and EfficientNetB1 (×20, 512) models falsely predicted adenocarcinoma on the marked blue-dots that pathologists marked when they performed diagnosis (Figure 3C,D). In contrast, both TL-colon poorly ADC-2 (×20, 512) and TL-lung carcinoma (×10, 512) models precisely predicted adenocarcinoma (Figure 3A,B). The model (TL-colon poorly ADC-2 (×20, 512)) achieved the highest ROC-AUC of 0.984 (CI: 0.956–1.000) and lowest log loss of 0.127 (CI: 0.076–0.205) for prostate adenocarcinoma classification in TUR-P hospital-A test sets and also achieved high ROC-AUCs in public dataset (0.947, CI: 0.922–0.972) and needle biopsy test sets (0.913, CI: 0.887–0.939) (Table 4). In all test sets, the model ((TL-colon poorly ADC-2 (×20, 512)) achieved very high accuracy (0.821–0.969), sensitivity (0.764–0.900), and specificity (0.884–0.992) (Table 5). As shown in Figure 2 and Figure 3 and Table 4 and Table 5, the model (TL-colon poorly ADC-2 (×20, 512)) is fully applicable for prostate adenocarcinoma classification in TUR-P WSIs as well as the TCGA public WSI dataset and even needle biopsy WSIs. Figure 4, Figure 5, Figure 6, Figure 7 and Figure 8 show representative WSIs of true positive, true negative, false positive, and false negative, respectively, from using the model (TL-colon poorly ADC-2 (×20, 512)).

### 3.3. True Positive Prostate Adenocarcinoma Prediction of TUR-P WSIS

Our model (TL-colon poorly ADC-2 (×20, 512)) satisfactorily predicted adenocarcinoma in TUR-P WSIs (Figure 4A,B). According to the histopathological report and additional pathologist’s review, in this WSI (Figure 4A), there were three tissue fragments (highlighted with yellow-triangles) with prostate adenocarcinoma cell infiltration (Figure 4C,E,F,H,I,K). The heatmap image (Figure 4B) shows true-postive predictions in these fragments (yellow-triangles) (Figure 4D,E,G,H,J,K) without false positive predictions in other tissue fragments that were histopathologically evaluated as nodular hyperplasia (benign prostatic hyperplasia) and without evidence of malignancy (Figure 4A,B). This not only occurred with the representative WSI (Figure 4), as our model (TL-colon poorly ADC-2 (×20, 512)) precisely predicted a wide variety of prostate adenocarcinoma histopathological features (Figure 5): medium-sized, discrete, and distinct neoplastic glands (Gleason pattern 3) (Figure 5A,B); medium-sized discrete and distinct glands with ill-formed glands (Gleason score 3 + 4) (Figure 5C,D); ill-formed glands (Gleason pattern 4) (Figure 5E,F); and cribriform pattern (Gleason pattern 4) (Figure 5G,H).

### 3.4. True Negative Prostate Adenocarcinoma Prediction of TUR-P WSIS

Our model (TL-colon poorly ADC-2 (×20, 512)) showed true negative predictions of prostate adenocarcinoma in TUR-P WSIs (Figure 6A,B). In Figure 6A, histopathologically, there was nodular hyperplasia (benign prostatic hyperplasia) with chronic inflammation in all tissue fragments without evidence of malignancy (Figure 6A,C–F), which were not predicted as prostate adenocarcinoma (Figure 6B,C,E).

### 3.5. False Positive Prostate Adenocarcinoma Prediction of TUR-P WSIS

According to the histopathological reports and additional pathologist’s review, there was no prostate adenocarcinoma observed in these TUR-P WSIs (Figure 7A,E,H). Our model (TL-colon poorly ADC-2 (×20, 512)) showed false positive predictions of prostate adenocarcinoma (Figure 7B–D,F,G,I,J). These false positive tissue areas (Figure 7B–D,F,G,I,J) showed xanthogranulomatous inflammation (Figure 7A,C,D), macrophagic infiltration (Figure 7E,G), and squamous metaplasia with pseudo-koilocytosis (Figure 7H,J), which could be the primary cause of false positives due to its morphological similarity in adenocarcinoma cells.

### 3.6. False Negative Prostate Adenocarcinoma Prediction of TUR-P WSIS

According to the histopathological report and additional pathologist’s review, in this TUR-P WSI (Figure 8A), there were a very small number of adenocarcinoma cells infiltrating in a tissue fragment (Figure 8C), and pathologists have marked them with blue-dots. However, our model (TL-colon poorly ADC-2 (×20, 512)) did not predict any prostate adenocarcinoma cells (Figure 8B,C).

## 4. Discussion

In this study, we trained deep learning models for the classification of prostate adenocarcinoma in TUR-P WSIs. Of the four models we trained (Table 4), the best model (TL-colon poorly ADC-2 (×20, 512)) achieved ROC-AUCs in the range of 0.896–0.984 on the TUR-P test set. The best model (TL-colon poorly ADC-2 (×20, 512)) also achieved high ROC-AUCs on needle biopsy (0.913) and TCGA public dataset (0.947) test set. The model (TL-lung carcinoma (×10, 512)) also achieved high ROC-AUCs in all test sets but was lower than the best one (TL-colon poorly ADC-2 (×20, 512)). The other two models were trained using the EfficientNetB1 [37] models starting with pre-trained weights on ImageNet at different magnifications (×10 and ×20) and tile sizes (224 × 224 px, 512 × 512 px). The models based on EfficientNetB1 (EfficientNetB1 (×10, 224) and EfficientNetB1 (×20, 512)) achieved robust high ROC-AUC values on TUR-P Hospital-B test sets compared to other test sets (Table 4). This shows that additional pre-training on other histopathological images was beneficial. Based on the prediction heatmap images of prostate adenocarcinoma, it was obvious that the models based on EfficientNetB1 (EfficientNetB1 (×10, 224) and EfficientNetB1 (×20, 512)) incorrectly predicted blue ink dots, which pathologists had marked during diagnosis, as prostate adenocarcinoma (Figure 3). Based on this finding, we have looked over WSIs in TUR-P Hospital-B test sets (Table 2) and most of adenocarcinoma positive WSIs (28 out of 30 WSIs) had ink dots on WSIs, which were falsely predicted as adenocarcinoma. On the other hand, transfer learning models (TL-colon poorly ADC-2 (×20, 512) and TL-lung carcinoma (×10, 512)) revealed no false positive predictions on ink dots (Figure 3); this is because those models had been trained on WSIs with ink labelled as non-neoplastic. The best model (TL-colon poorly ADC-2 (×20, 512)) and the second best model (TL-lung carcinoma (×10, 512)) were trained by the transfer learning approach from our existing colon poorly differentiated adenocarcinoma classification model [31] and lung carcinoma classification model [45] based on the findings of ROC-AUC and log loss values on TUR-P test sets (TUR-P Hospital-A-B) using existing adenocarcinoma classification models (Table 3). We used the partial finetuning approach [36] to train the models faster, as there are less weights involved in tuning. We used only 1020 TUR-P WSIs (adenocarcinoma: 79 WSIs; benign: 941 WSIs) (Table 1) without manual annotations by pathologists [28,34,44]. We see that by specifically training on TUR-P WSIs, the models significantly improved prediction performances on the TUR-P test set (Table 4) compared to a previous study [35] that had lower ROC-AUC (0.737–0.909) and higher log loss (3.269–4.672) values. The combination of both models can provide accurate prostate adenocarcinoma classification on both needle biopsy [35] and TUR-P WSIs in routine histopathological diagnostic workflow.

Nodular hyperplasia (benign prostatic hyperplasia) is a common benign disorder of the prostate as a histopathological diagnosis referring to the nodular enlargement of the gland caused by hyperplasia in both glandular and stromal components within the prostatic transition zone and results in varying degrees of urinary obstruction, which sometimes requiring surgical interventions, including TUR-P [46]. Importantly, incidental prostate cancers are diagnosed at the time of TUR-P for benign prostatic disease [10]. According to the literature search, cancers, particularly prostate adenocarcinoma, are detected incidentally at around 5–17% of TUR-P specimens [4,5,6,7,8,9,10], meaning that around 83–95% of TUR-P specimens are benign lesions, which is nearly identical to the ratio of adenocarcinoma in the TUR-P test sets (Table 2). Therefore, the high values of specificity (0.884–0.992) in the best model are noteworthy (Table 5). Moreover, heatmap images revealed true negative predictions perfectly on each non-neoplastic fragment in both adenocarcinoma (Figure 4) and benign (non-neoplastic) (Figure 6) WSIs. Thus, heatmap images predicted by the best model would provide great benefits for pathologists who have to report the detail descriptions of many TUR-P specimens in routine clinical practices.

One limitation of this study is that it primarily included specimens from a limited number of hospitals and suppliers in Japan; therefore, the model could potentially be biased to such specimens. Further validations on a wide variety of specimens from multiple different origins would be essential for ensuring the robustness of the model. Another potential validation study could involve the comparison of the performance of the model against pathologists in a clinical setting. Another limitation of the study is that it simply performed classifications with respect to adenocarcinoma regardless of the Gleason score; in clinical practices, being able to classify the Gleason score would be of more interest.

## 5. Conclusions

The best deep learning model established in the present study offers promising results that indicate that it could be beneficial as a screening aid for pathologists prior to observing histopathology on glass slides or WSIs. At the same time, the model could be used as a double-checking tool for reducing the risk of missed cancer foci (incidental adenocarcinoma in TUR-P specimens). The most important advantage of using a fully automated computational tool is that it can systematically handle large amounts of WSIs without potential bias due to the fatigue commonly experienced by pathologists, which could drastically alleviate the heavy clinical burden of practical pathology diagnoses when using conventional microscopes.

## Figures and Tables

**Figure 1 cancers-14-04744-f001:**
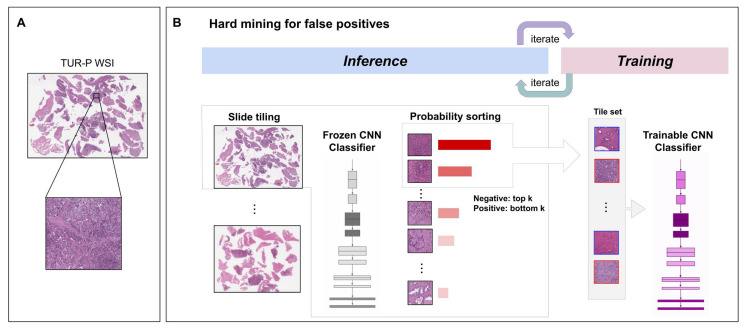
Schematic diagrams of training method overview. (**A**) shows a representative transurethral resection of the prostate (TUR-P) whole-slide image (WSI) (65,736 × 47,326 px) with a zoomed-in tile (224 × 224 px). (**B**) During training, we iteratively alternated between inference and training. During the inference step, model weights were frozen, and the model was used to select tiles with the highest probability after applying it on entire tissue regions of each WSI. The top k tiles with the highest probabilities were then selected from each WSI and placed into a queue. During training, the selected tiles from multiple WSIs formed a training batch and were used to train the model.

**Figure 2 cancers-14-04744-f002:**
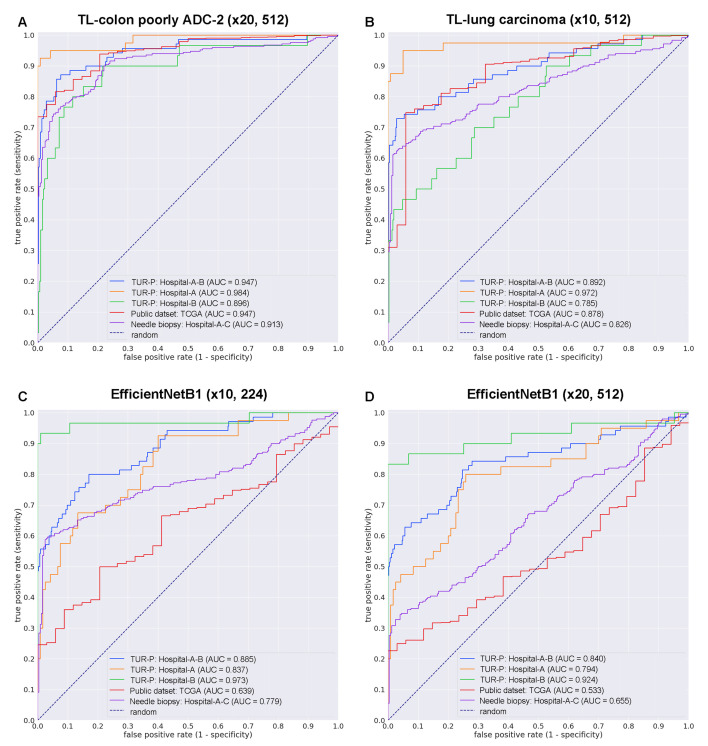
ROC curves with AUCs from four different trained deep learning models (**A**–**D**) on the test sets: (**A**) transfer learning (TL) model from an existing colon poorly differentiated adenocarcinoma (ADC) classification model with a tile size of 224 px and magnification at ×20; (**B**) TL model from an existing lung carcinoma classification model with a tile size of 512 px and magnification at ×10; (**C**) EfficientNetB1 model with a tile size of 224 px and magnification at ×10; and (**D**) EfficientNetB1 model with a tile size of 512 px and magnification at ×20.

**Figure 3 cancers-14-04744-f003:**
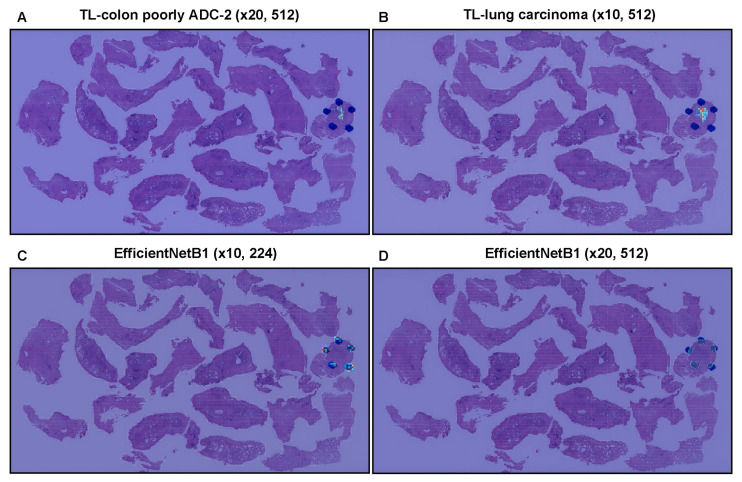
Comparison of adenocarcinoma predictions in the transurethral resection of the prostate (TUR-P) whole-slide image (WSI) of four trained deep learning models (**A**–**D**). In transfer learning (TL) models from colon poorly differentiated adenocarcinoma (**A**) and lung carcinoma (**B**), heatmap images show a true-postive prediction of adenocarcinoma where pathologists marked the surrounding with blue-dots. In EfficientNetB1 models (**C**,**D**), heatmap images show a false positive prediction of adenocarcinoma on the marked blue-dots. The heatmap uses the jet color map where blue indicates low probability and red indicates high probability.

**Figure 4 cancers-14-04744-f004:**
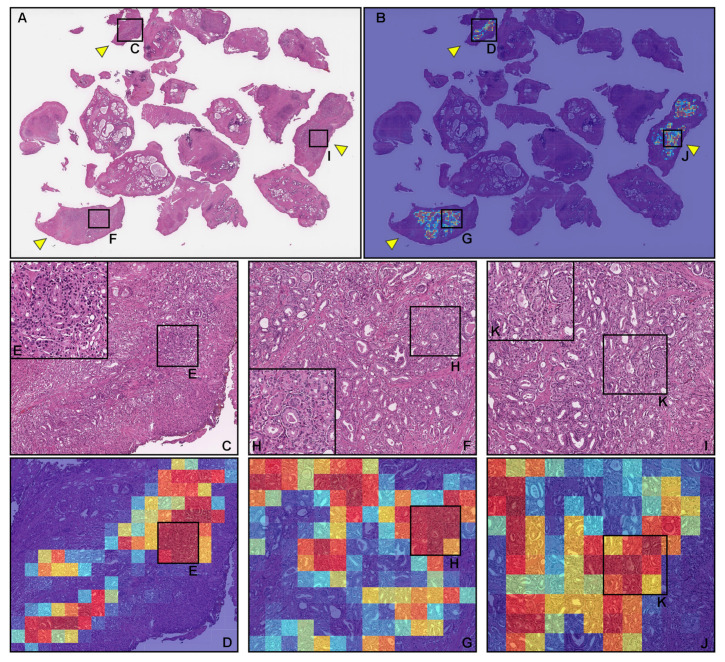
A representative example of prostate adenocarcinoma true positive prediction outputs on a whole-slide image (WSI) from the transurethral resection of the prostate (TUR-P) test sets using the model (TL-colon poorly ADC-2 (×20, 512)). In the prostate adenocarcinoma WSI of TUR-P specimen (**A**), according to the histopathological diagnostic report, adenocarcinoma cells that were infiltrated in the three tissue fragments are highlighted with yellow-triangles. The heatmap image (**B**) shows true-postive predictions of prostate adenocarcinoma cells (**D**,**G**,**J**), which correspond, respectively, to H&E histopathology (**C**–**K**). The heatmap image (**B**) also shows no positive predictions (true negative predictions) in tissue fragments, without evidence of adenocarcinoma infiltration (**A**). The heatmap uses the jet color map in which blue indicates low probability and red indicates high probability.

**Figure 5 cancers-14-04744-f005:**
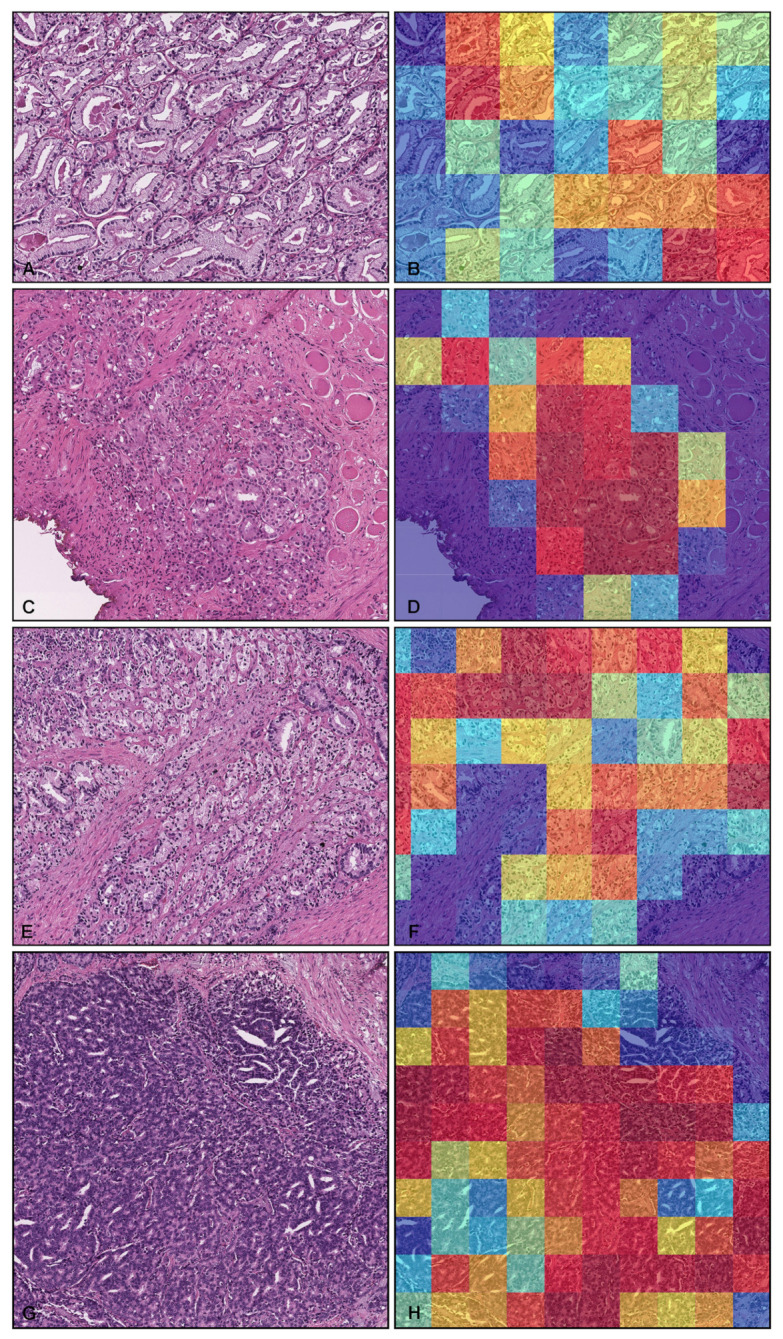
Representative histopathological examples of prostate adenocarcinoma true positive prediction outputs on whole-slide images (WSIs) from transurethral resection of the prostate (TUR-P) test sets using the model (TL-colon poorly ADC-2 (×20, 512)). Depiction of prostate adenocarcinoma histopathologies and corresponding heatmap images of adenocarcinoma prediction outputs: (**A**,**B**) medium-sized, discrete, and distinct neoplastic glands (Gleason pattern 3); (**C**,**D**) medium-sized discrete and distinct glands with ill-formed glands (Gleason score 3 + 4); (**E**,**F**) ill-formed glands (Gleason pattern 4); (**G**,**H**) cribriform pattern (Gleason pattern 4). The heatmap uses the jet color map where blue indicates low probability and red indicates high probability.

**Figure 6 cancers-14-04744-f006:**
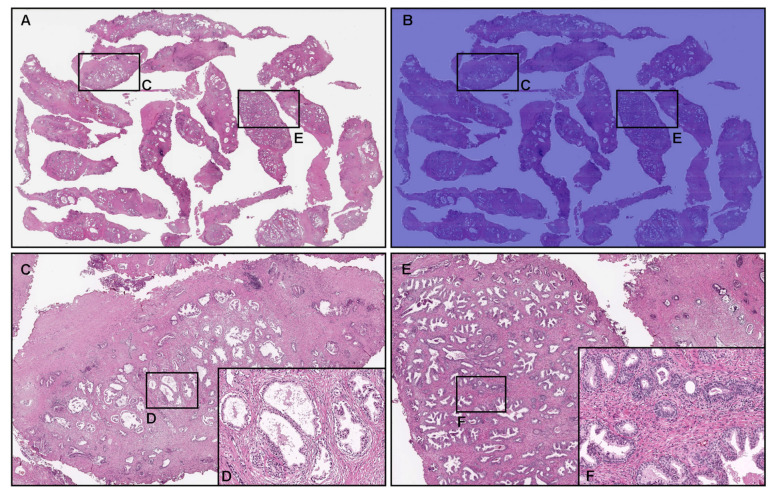
Representative true negative prostate adenocarcinoma prediction outputs on a whole-slide image (WSI) from transurethral resection of the prostate (TUR-P) test sets using the model (TL-colon poorly ADC-2 (×20, 512)). Histopathologically, in (**A**), there was nodular hyperplasia (benign prostatic hyperplasia) with chronic inflammation without any evidence of malignancy (**C**–**F**). The heatmap image (**B**,**C**,**E**) shows a true negative prediction of prostate adenocarcinoma. The heatmap uses the jet color map where blue indicates low probability and red indicates high probability.

**Figure 7 cancers-14-04744-f007:**
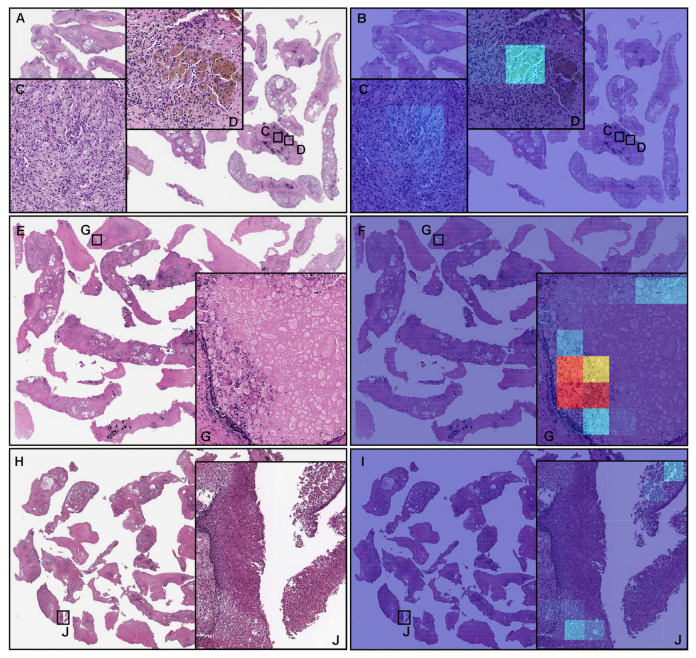
Representative examples of prostate adenocarcinoma false positive prediction outputs on whole-slide images (WSIs) from the transurethral resection of the prostate (TUR-P) test sets using the model (TL-colon poorly ADC-2 (×20, 512)). Histopathologically, (**A**,**E**,**H**) have no evidence of adenocarcinoma infiltration. The heatmap images (**B**,**F**,**I**) exhibit false positive predictions of prostate adenocarcinoma (**C**,**D**,**G**,**J**) where the tissues consist of xanthogranulomatous inflammation (**C**,**D**), macrophagic infiltration (**G**), and squamous metaplasia with pseudo-koilocytosis (**J**), which are most likely the primary cause of the false positive prediction due to its morphological similarity to prostate adenocarcinoma cells. The heatmap uses the jet color map where blue indicates low probability and red indicates high probability.

**Figure 8 cancers-14-04744-f008:**
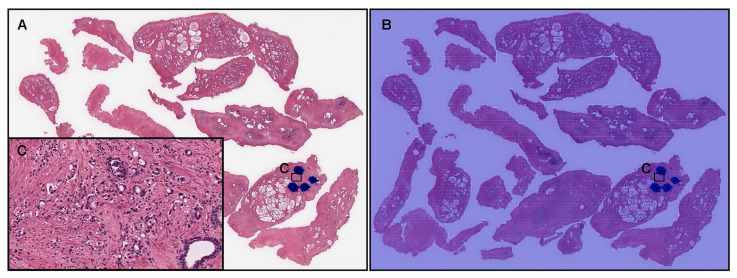
A representative example of prostate adenocarcinoma false negative prediction outputs on a whole-slide image (WSI) from the transurethral resection of the prostate (TUR-P) test sets using the model (TL-colon poorly ADC-2 (×20, 512)). According to the histopathological diagnostic report, this case (**A**) has a very small number of adenocarcinoma foci (cells) in (**C**) where pathologists marked surrounding with blue-dots but not in other areas that consist of nodular hyperplasia. The heatmap image (**B**) exhibited no positive adenocarcinoma prediction (**C**). The heatmap uses the jet color map where blue indicates low probability and red indicates high probability.

**Table 1 cancers-14-04744-t001:** Distribution of the transuretheral resection of prostate (TUR-P) whole-slide images (WSIs) in training and validation sets obtained from two hospitals (A and B).

		Adenocarcinoma	Benign	Total
Training set	Hospital-A	59	222	281
	Hospital-B	20	719	739
Validation set	Hospital-A	10	10	20
	Hospital-B	10	10	20
	total	99	961	1060

**Table 2 cancers-14-04744-t002:** Distribution of whole-slide images (WSIs) in the transuretheral resection of the prostate (TUR-P), public dataset (TCGA), and needle biopsy test sets obtained from three hospitals (A–C).

		Adenocarcinoma	Benign	Total
TUR-P	Hospital-A–B	70	430	500
	Hospital-A	40	120	160
	Hospital-B	30	310	340
Public dataset	TCGA	733	34	767
Needle biopsy	Hospital-A–C	250	250	500

**Table 3 cancers-14-04744-t003:** ROC-AUC and log loss results for adenocarcinoma classification on transuretheral resection of prostate (TUR-P) test sets (Hospital-A-B) using existing adenocarcinoma classification models.

Existing Models	ROC-AUC	Log Loss
Breast IDC (×10, 512)	0.737 [0.664–0.807]	1.428 [1.340–1.530]
Breast IDC, DCIS (×10, 224)	0.635 [0.565–0.720]	3.783 [3.624–3.929]
Colon ADC, AD (×10, 512)	0.608 [0.546–0.679]	3.812 [3.595–4.028]
Colon poorly ADC-1 (×20, 512)	0.780 [0.713–0.840]	0.863 [0.811–0.913]
Colon poorly ADC-2 (×20, 512)	0.771 [0.681–0.837]	0.859 [0.890–0.914]
Stomach ADC, AD (×10, 512)	0.762 [0.689–0.833]	3.133 [2.948–3.268]
Stomach poorly ADC (×20, 224)	0.617 [0.529–0.698]	1.588 [1.504–1.657]
Stomach SRCC (×10, 224)	0.670 [0.600–0.734]	0.549 [0.499–0.606]
Pancreas EUS-FNA ADC (×10, 224)	0.808 [0.746–0.888]	1.080 [1.031–1.142]
Lung Carcinoma (×10, 512)	0.737 [0.662–0.801]	0.357 [0.298–0.423]

**Table 4 cancers-14-04744-t004:** ROC-AUC and log loss results for adenocarcinoma classification on the transuretheral resection of the prostate (TUR-P), public dataset (TCGA), and needle biopsy test sets using trained models.

		TL-colon poorly ADC-2 (×20, 512)	
		ROC-AUC	Log-loss
TUR-P	Hospital-A–B	0.947 [0.910–0.976]	0.191 [0.146–0.242]
	Hospital-A	0.984 [0.956–1.000]	0.127 [0.076–0.205]
	Hospital-B	0.896 [0.822–0.956]	0.221 [0.160–0.299]
Public dataset	TCGA	0.947 [0.922–0.972]	0.335 [0.288–0.390]
Needle biopsy	Hospital-A–C	0.913 [0.887–0.939]	0.587 [0.480–0.700]
		TL-lung carcinoma (×10, 512)	
		ROC-AUC	Log-loss
TUR-P	Hospital-A–B	0.892 [0.860–0.948]	0.328 [0.282–0.364]
	Hospital-A	0.972 [0.917–0.998]	0.277 [0.217–0.364]
	Hospital-B	0.785 [0.688–0.870]	0.351 [0.301–0.403]
Public dataset	TCGA	0.878 [0.822–0.929]	0.258 [0.213–0.299]
Needle biopsy	Hospital-A–C	0.826 [0.786–0.860]	0.808 [0.702–0.931]
		EfficientNetB1 (×10, 224)	
		ROC-AUC	Log-loss
TUR-P	Hospital-A–B	0.885 [0.829–0.927]	0.239 [0.181–0.298]
	Hospital-A	0.837 [0.752–0.909]	0.479 [0.318–0.619]
	Hospital-B	0.973 [0.916–1.000]	0.126 [0.092–0.168]
Public dataset	TCGA	0.639 [0.563–0.716]	3.800 [3.613–3.977]
Needle biopsy	Hospital-A–C	0.779 [0.736–0.822]	0.659 [0.552–0.769]
		EfficientNetB1 (×20, 512)	
		ROC-AUC	Log-loss
TUR-P	Hospital-A–B	0.840 [0.767–0.897]	0.315 [0.269–0.377]
	Hospital-A	0.794 [0.681–0.872]	0.451 [0.323–0.601]
	Hospital-B	0.924 [0.856–0.998]	0.251 [0.203–0.290]
Public dataset	TCGA	0.533 [0.464–0.616]	2.611 [2.485–2.721]
Needle biopsy	Hospital-A–C	0.655 [0.609–0.702]	1.785 [1.551–1.956]

**Table 5 cancers-14-04744-t005:** Scores of accuracy, sensitivity, specificity, Negative Predictive Value (NPV), and Positive Predictive Value (PPV) on the transuretheral resection of the prostate (TUR-P), public dataset (TCGA), and needle biopsy test sets using the best model (TL-colon poorly ADC-2 (×20, 512)).

		Accuracy	Sensitivity	Specificity
TUR-P	Hospital-A–B	0.916 [0.892–0.938]	0.871 [0.794–0.951]	0.923 [0.897–0.945]
	Hospital-A	0.969 [0.938–0.994]	0.900 [0.800–0.976]	0.992 [0.974–1.000]
	Hospital-B	0.874 [0.841–0.909]	0.767 [0.613–0.906]	0.884 [0.848–0.922]
Public dataset	TCGA	0.821 [0.793–0.849]	0.816 [0.786–0.843]	0.941 [0.852–1.000]
Needle biopsy	Hospital-A–C	0.844 [0.812–0.874]	0.764 [0.710–0.813]	0.924 [0.886–0.956]
		NPV	PPV	
TUR-P	Hospital-A–B	0.978 [0.963–0.993]	0.649 [0.550–0.730]	
	Hospital-A	0.968 [0.928–0.992]	0.973 [0.909–1.000]	
	Hospital-B	0.975 [0.955–0.990]	0.390 [0.275–0.537]	
Public dataset	TCGA	0.192 [0.132–0.258]	0.997 [0.992–1.000]	
Needle biopsy	Hospital-A–C	0.797 [0.750–0.840]	0.910 [0.865–0.946]	

## Data Availability

The datasets generated during and/or analysed during the current study are not publicly available due to specific institutional requirements governing privacy protection but are available from the corresponding author upon reasonable request. The datasets that support the findings of this study are available from Kamachi Group Hospitals (Fukuoka, Japan), but restrictions apply to the availability of these data, which were used under a data-use agreement that was made according to the Ethical Guidelines for Medical and Health Research Involving Human Subjects as set by the Japanese Ministry of Health, Labour and Welfare (Tokyo, Japan) and, thus, are not publicly available. However, the data are available from the authors upon reasonable request for private viewing and with permission from the corresponding medical institutions within the terms of the data use agreement and if compliant with the ethical and legal requirements as stipulated by the Japanese Ministry of Health, Labour and Welfare.

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
