# Peer review of "Transfer Learning for Adenocarcinoma Classifications in the Transurethral Resection of Prostate Whole-Slide Images"

_cancers, 2022, doi:10.3390/cancers14194744_

Round 1

Reviewer 1 Report

Thank you for the opportunity to review the manuscript entitled, "Transfer learning for adenocarcinoma classification in transurethral resection of the prostate whole slide images." The authors train deep learning models to classify transurethral resection of the prostate (TUR-P) whole slide images as either cancer or benign using transfer and weakly supervised learning. The authors found that their models received ROC-AUS up to 0.984. The clinical topic is essential, as better detection of prostate histology is a growing field in prostate cancer, the most common cancer in men. However, I have several comments to improve the quality of the manuscript.

  1. I don't understand the sentence, "Since diagnosing a large number of cases containing TUR-P specimens which are characterized by a very large volume of tissue fragments by pathologists using a conventional microscope is time-consuming and limited in terms of human resources ". 
  2. The abstract can be significantly improved. The authors spend the first 10-11 sentences in the abstract focusing on the study's background and objective. It would be much more beneficial to the author team if they spent 1-3 sentences max on the background, 1-2 on the objective, and then spent the rest of the space adding information in the methods and results sections. My biggest concern is the lack of an overview of data and methods. How much data was trained and tested? Which years is the data from?  What data do the authors work with_ What is novel about the deep learning models? What does the data look like? What preprocessing happened? 
  3. It will benefit the paper if the authors report sensitivity, specificity, NPV, and PPV. 
  4. An AUC of 0.984 is almost perfect. In nature, data is often not 100% perfect, so seeing an AUC close to 1 could mean different things. Maybe the authors are testing on training data, or perhaps the evaluation code is slightly imprecise.
  5. It would benefit the paper if the authors released and shared their code online so people could reproduce the findings. Possibly GitHub or similar pages. 
  6. Did the authors try to classify all cancer or clinically significant cancer? Clinically significant cancer (Gleason score seven or higher) is likely much more interested in a clinical readership than all cancer.
  7. Please add additional limitations to the study. The authors could touch on possible weaknesses in labeling.
  8. Could the authors do a 5-fold cross-validation?
  9. At the end of the introduction, results and details about methods can be left out. Keeping the objective is fine and enough. 
  10. How many patients were the 2,060 histopathology slides from?

Author Response

Reviewer 1:

Thank you for the opportunity to review the manuscript entitled, "Transfer learning for adenocarcinoma classification in transurethral resection of the prostate whole slide images." The authors train deep learning models to classify transurethral resection of the prostate (TUR-P) whole slide images as either cancer or benign using transfer and weakly supervised learning. The authors found that their models received ROC-AUS up to 0.984. The clinical topic is essential, as better detection of prostate histology is a growing field in prostate cancer, the most common cancer in men. However, I have several comments to improve the quality of the manuscript.

I don't understand the sentence, "Since diagnosing a large number of cases containing TUR-P specimens which are characterized by a very large volume of tissue fragments by pathologists using a conventional microscope is time-consuming and limited in terms of human resources ". 

Response: We’ve clarified the sentences and changed it to:

TUR-P specimens contain a large number of fragmented prostate tissues; this makes it time-consuming and a waste of human resources for pathologists to check fragmented prostate tissues one by one.

The abstract can be significantly improved. The authors spend the first 10-11 sentences in the abstract focusing on the study's background and objective. It would be much more beneficial to the author team if they spent 1-3 sentences max on the background, 1-2 on the objective, and then spent the rest of the space adding information in the methods and results sections. 

Response: We’ve shortened the background part.

My biggest concern is the lack of an overview of data and methods. How much data was trained and tested? Which years is the data from?  What data do the authors work with_ What is novel about the deep learning models? What does the data look like? What preprocessing happened?

Response: 

  • How much data was trained and tested?

Tables 1 and 2 contain a summary of the exact number of WSIs used for training and testing, respectively. 

  • Which years is the data from?

We’ve added a sentence in the data section to clarify that the data is from 2017-2019.

  • What data do the authors work with?

The “materials and method” section describe what type of data we are working with, in particular, the “Clinical cases and pathological records”. In there, amongst other details, we state that we are using  “a total of 2,060 H&E (hematoxylin & eosin) stained histopathological specimen slides of human prostate adenocarcinoma and benign (non-neoplastic) lesions --1,560 TUR-P and 500 needle biopsy”. 

  • What does the data look like?

The data is just the typical specimens obtained from hospitals during routine clinical practice. Figures 3-8 show examples of different specimens that are part of the data. We hope they illustrate what the data looks like.

  • What is novel about the deep learning models?

There is no novelty with the deep learning models. It is a clinical application study, and the journal accepts clinical applications.

  • What preprocessing happened?

In the “deep learning models” section you can find details about what preprocessing was performed on the images.

It will benefit the paper if the authors report sensitivity, specificity, NPV, and PPV.

Response: We already report the sensitivity and specificity in table 5. We’ve extended the table to include NPV and PPV.

An AUC of 0.984 is almost perfect. In nature, data is often not 100% perfect, so seeing an AUC close to 1 could mean different things. Maybe the authors are testing on training data, or perhaps the evaluation code is slightly imprecise.

Response: We did not test on the training data. If you have a look at Table 4, there’s a wide range of AUC scores and confidence intervals ranging from 0.63 up to 0.984. 0.984 was just the highest achieved AUC on the TUR-P Hospital A test set using model TL-colon poorly ADC-2 (x20, 512). You can also note the Hospital A test set has about 40 Adenocarcinoma cases and 120 benign.

It would benefit the paper if the authors released and shared their code online so people could reproduce the findings. Possibly GitHub or similar pages. 

Response: We did not develop any novel methodology and simply applied existing methods for a clinical application. It is not a requirement of the journal to share code.

https://www.mdpi.com/journal/cancers/instructions#suppmaterials

“For work where novel computer code was developed, authors should release the code either by depositing in a recognized, public repository such as GitHub or uploading as supplementary information to the publication. The name, version, corporation and location information for all software used should be clearly indicated. Please include all the parameters used to run software/programs analyses.”

Did the authors try to classify all cancer or clinically significant cancer? Clinically significant cancer (Gleason score seven or higher) is likely much more interested in a clinical readership than all cancer.

Response: We classified all cancer (all adenocarcinoma) regardless of Gleason score.

Please add additional limitations to the study. The authors could touch on possible weaknesses in labeling.

Response: We have added a limitation that we simply classify all cancers regardless of Gleason score.

Could the authors do a 5-fold cross-validation?

Response: We performed bootstrapping [1] instead of cross validation to obtain the confidence interval estimate. Bootstrapping is an alternative method to cross validation, and is also commonly used. References [2,3] are two publications in high impact journals (Impact factor > 40) that deal with a similar application and do not perform cross validation, and instead use the bootstrapping method.

In addition, the goal of using bootstrapping or cross validation is to obtain an estimate of how well the model would perform on a new test set that is potentially different from data used during training. In our case we have done that using the TCGA and core needle biopsy test sets (see Table 2) which are not the same as TUR-P used for training.

Our main argument was that we have an extremely large dataset with millions of tiles extracted from the WSIs. For reference we provide a video from Andrew Ng, a top researcher and educator in machine learning from Stanford University about the construction of training/dev/test for deep learning https://youtu.be/1waHlpKiNyY?t=247

[1] Efron, B.; Tibshirani, R.J. An introduction to the bootstrap; CRC press, 1994

[2] Bulten, Wouter, et al. "Automated deep-learning system for Gleason grading of prostate cancer using biopsies: a diagnostic study." The Lancet Oncology 21.2 (2020): 233-241.

[3] Campanella, Gabriele, et al. "Clinical-grade computational pathology using weakly supervised deep learning on whole slide images." Nature medicine 25.8 (2019): 1301-1309. https://www.nature.com/articles/s41591-019-0508-1

At the end of the introduction, results and details about methods can be left out. Keeping the objective is fine and enough.

Response: We do not go much into detail and simply provide a summary for what the reader will expect.

How many patients were the 2,060 histopathology slides from?

Response: We have added a sentence in the materials and methods section that clarifies the number of patients.

1560 TUR-P WSIs → 276 cases (patients); 500 needle biopsy WSIs → 238 cases (patients)

Reviewer 2 Report

This is a well-written manuscript, Masayuki Tsuneki and colleagues showed that deep learning models for the classification of prostate adenocarcinoma in TUR-P WSIs, and they found the best model (TL-colon poorly ADC-2 (x20, 512)) achieved ROC-AUCs in the range of 0.896 - 0.984 on the TUR-P test sets. This model also achieved high ROC-AUCs on needle biopsy (0.913) and TCGA public dataset (0.947) test sets. As usual, there was false positive and false negative detection by the algorithm. 

A major point:

1.       More hospital cases for training and validation in comparison to the number of samples used in the test cases. This should be addressed.  

2.       A minor issue: The language of the manuscript could be improved by a professional native English language editor.

Author Response

Reviewer 2:

This is a well-written manuscript, Masayuki Tsuneki and colleagues showed that deep learning models for the classification of prostate adenocarcinoma in TUR-P WSIs, and they found the best model (TL-colon poorly ADC-2 (x20, 512)) achieved ROC-AUCs in the range of 0.896 - 0.984 on the TUR-P test sets. This model also achieved high ROC-AUCs on needle biopsy (0.913) and TCGA public dataset (0.947) test sets. As usual, there was false positive and false negative detection by the algorithm. 

A major point:

  1.       More hospital cases for training and validation in comparison to the number of samples used in the test cases. This should be addressed.  

Response: This is the standard approach in machine learning where more cases are used for training compared to testing.

  1.       A minor issue: The language of the manuscript could be improved by a professional native English language editor.

Response: We have done another pass on the manuscript.

Reviewer 3 Report

Introduction: Excellent problem description and significance of the work.

Methods: Clear, concise,  and appropriate to the study and audience. Two notes for improvement:

1. In Section 2.1, it would be helpful to understand the type of diagnostic agreement among the reviewer pathologists (lines 78-79). For example, did pathologists have to reach consensus agreement independent of each other or could they discuss their findings together? Was it diagnostic agreement as in "presence of adenocarcinoma" or did they also need to agree on the exact Gleason scoring?

2. For Figure 1: In order to help the Digital Pathology-novice reader understand the scale of a WSI image, it would be helpful to include a scale bar or other size indicator (ex. pixel size of WSI vs ROI).

Results: Logically presented with very good supporting figures and tables. The captions are very well done. The text description would strongly benefit from an English-editor to improve clarity and readability.

1. Figure 3: Due to the coloration and scale of the image, this figure would benefit from either cropping the image to the area of interest or creating insets of the area of interest at a larger scale (Like Figure 4. C-E; F-H; I-K) to better show the TP/FP areas.

2. Figure 6: It would benefit the authorsif the authors showed zoomed in areas of the heatmap image corresponding to the zoomed-in WSI H&E images so readers can easily see the lack of positive coloration in these areas (similar to Figure 7). 

Discussion: The discussion is an excellent summary of the work, including limitations and setting this work in the larger clinical framework.

Author Response

Reviewer 3:

Introduction: Excellent problem description and significance of the work.

Response: Thank you.

Methods: Clear, concise,  and appropriate to the study and audience. Two notes for improvement:

  1. In Section 2.1, it would be helpful to understand the type of diagnostic agreement among the reviewer pathologists (lines 78-79). For example, did pathologists have to reach consensus agreement independent of each other or could they discuss their findings together? Was it diagnostic agreement as in "presence of adenocarcinoma" or did they also need to agree on the exact Gleason scoring?

Response: It was a diagnostic agreement in “presence of adenocarcinoma”. We’ve added a clarification in the methods section.

  1. For Figure 1: In order to help the Digital Pathology-novice reader understand the scale of a WSI image, it would be helpful to include a scale bar or other size indicator (ex. pixel size of WSI vs ROI).

Response: We have added in the caption the actual size of the image (65,736 x 47,326 px) and ROI (224x224px) in pixels

Results: Logically presented with very good supporting figures and tables. The captions are very well done. The text description would strongly benefit from an English-editor to improve clarity and readability.

Response: Thanks. We have done another proofreading pass on the paper.

  1. Figure 3: Due to the coloration and scale of the image, this figure would benefit from either cropping the image to the area of interest or creating insets of the area of interest at a larger scale (Like Figure 4. C-E; F-H; I-K) to better show the TP/FP areas.

Response: The aim of Figure 3 was to show a full view of the WSI. Figures 4-8 show other examples with ROIs. If the paper is published online, the readers should be able to zoom in to view the full resolution image.

  1. Figure 6: It would benefit the authorsif the authors showed zoomed in areas of the heatmap image corresponding to the zoomed-in WSI H&E images so readers can easily see the lack of positive coloration in these areas (similar to Figure 7). 

Response: Figure 6 shows a true negative case, so by virtue of being a true negative all the coloration will be blue, which is why we don’t show an enlarged ROI and it would be redundant with C and D. We therefore took the opportunity to simply show the tissue regions in the ROIs in C and D.

Discussion: The discussion is an excellent summary of the work, including limitations and setting this work in the larger clinical framework.

Response: Thank you.

Round 2

Reviewer 1 Report

I believe the manuscript has been sufficiently improved to warrant publication in Cancers. Most my comments and concerns have been adequately addressed.